

# Bayesian computation for the common coefficient of variation of delta-lognormal distributions with application to common rainfall dispersion in Thailand

Noppadon Yosboonruang, Sa-Aat Niwitpong and Suparat Niwitpong

Department of Applied Statistics, King Mongkut's University of Technology North Bangkok, Bangkok, Thailand

## ABSTRACT

Rainfall fluctuation makes precipitation and flood prediction difficult. The coefficient of variation can be used to measure rainfall dispersion to produce information for predicting future rainfall, thereby mitigating future disasters. Rainfall data usually consist of positive and true zero values that correspond to a delta-lognormal distribution. Therefore, the coefficient of variation of delta-lognormal distribution is appropriate to measure the rainfall dispersion more than lognormal distribution. In particular, the measurement of the dispersion of precipitation from several areas can be determined by measuring the common coefficient of variation in the rainfall from those areas together. Herein, we compose confidence intervals for the common coefficient of variation of delta-lognormal distributions by employing the fiducial generalized confidence interval, equal-tailed Bayesian credible intervals incorporating the independent Jeffreys or uniform priors, and the method of variance estimates recovery. A combination of the coverage probabilities and expected lengths of the proposed methods obtained *via* a Monte Carlo simulation study were used to compare their performances. The results show that the equal-tailed Bayesian based on the independent Jeffreys prior was suitable. In addition, it can be used the equal-tailed Bayesian based on the uniform prior as an alternative. The efficacies of the proposed confidence intervals are demonstrated *via* applying them to analyze daily rainfall datasets from Nan, Thailand.

# INTRODUCTION

Currently, the anthropomorphic emissions of greenhouse gases, sulfate aerosols, and black carbon are having a seriously deleterious effect on the Earth's climate (*Nema, Nema & Roy, 2012*). This phenomenon is directly increasing the global temperature, warming the oceans, and melting the polar ice caps, thereby causing a rise in sea level and initiating extreme weather events (*NASA, 2020*). Southeast Asia is a tropical area that is affected by ocean currents, prevailing winds, and abundant rainfall during the monsoon season

Corresponding author
Sa-Aat Niwitpong,
sa-aat.n@sci.kmutnb.ac.th

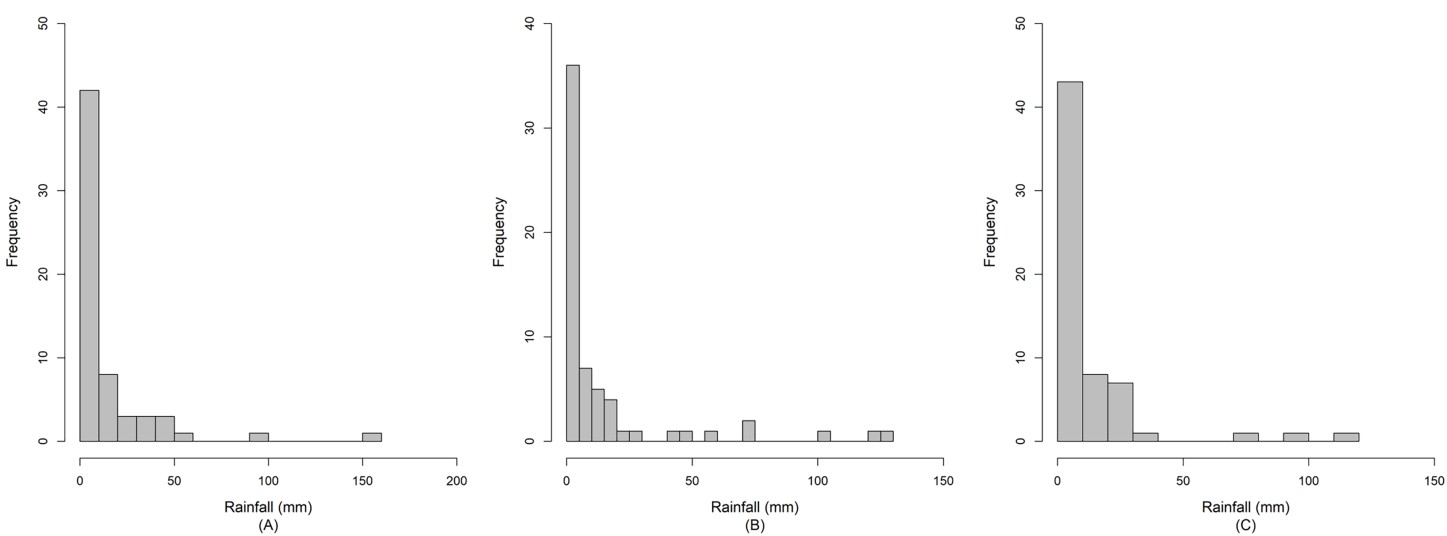

**Figure 1 Histograms of the daily rainfall data from (A) Chiang Klang, (B) Tha Wang Pha, and (C) Pua in Nan, Thailand.**

(*WorldAtlas, 2021*). Thailand is located in Southeast Asia, where the climate is influenced by the monsoon winds. Especially, the combined effect of the southwest monsoon, the Inter-Tropical Convergence Zone, and tropical cyclones causes plenty of rain to fall over the country (*Thai Meteorological Department, 2015*). Large amounts of rainfall cause regular flooding in some areas of the country, thereby leading to damage to property and loss of life. Moreover, Thailand is an agricultural country, and rainfall fluctuation makes it difficult to predict heavy precipitation that may cause loss of or damage to crops. Therefore, it is necessary to measure the dispersion of rainfall in specific areas by using statistical tools such as the coefficient of variation (CV) to enable accurate prediction of future catastrophic events. Nan is a province in Thailand located near the origin of the Nan River that flows into the Chao Phraya River. Furthermore, throughout the year, the precipitation in Nan fluctuates between a precipitation deficit and heavy rainfall. The latter accompanied by thunderstorms occurs in the late summer period, and due to the southwest monsoon, the amount of daily rainfall increases from mid-May to early October with the highest daily rainfall frequently in August or September, which can cause flooding in some areas (*Thai Meteorological Department, 2015*). Therefore, datasets of daily rainfall from the three areas (Chiang Klang, Tha Wang Pha, and Pua) in Nan province in August 2018 and 2019 were selected. These data comprise positive values that conform to a lognormal distribution, and true zero values, in which the frequency conforms to a binomial distribution, as presented in Fig. 1. In addition, the normality plots shown in Fig. 2 together with the Akaike information criterion (AIC) and the Bayesian information criterion (BIC) indicate that the daily rainfall data from the three areas follow delta-lognormal distributions. Furthermore, many researchers have reported that rainfall data follow a delta-lognormal distribution (*Fukuchi, 1988*; *Shimizu, 1993*; *Yue, 2000*; *Kong et al., 2012*; *Maneerat, Niwitpong & Niwitpong, 2019a, 2020a, 2020b*; *Yosboonruang, Niwitpong & Niwitpong, 2019b, 2020*; *Yosboonruang & Niwitpong, 2020*).
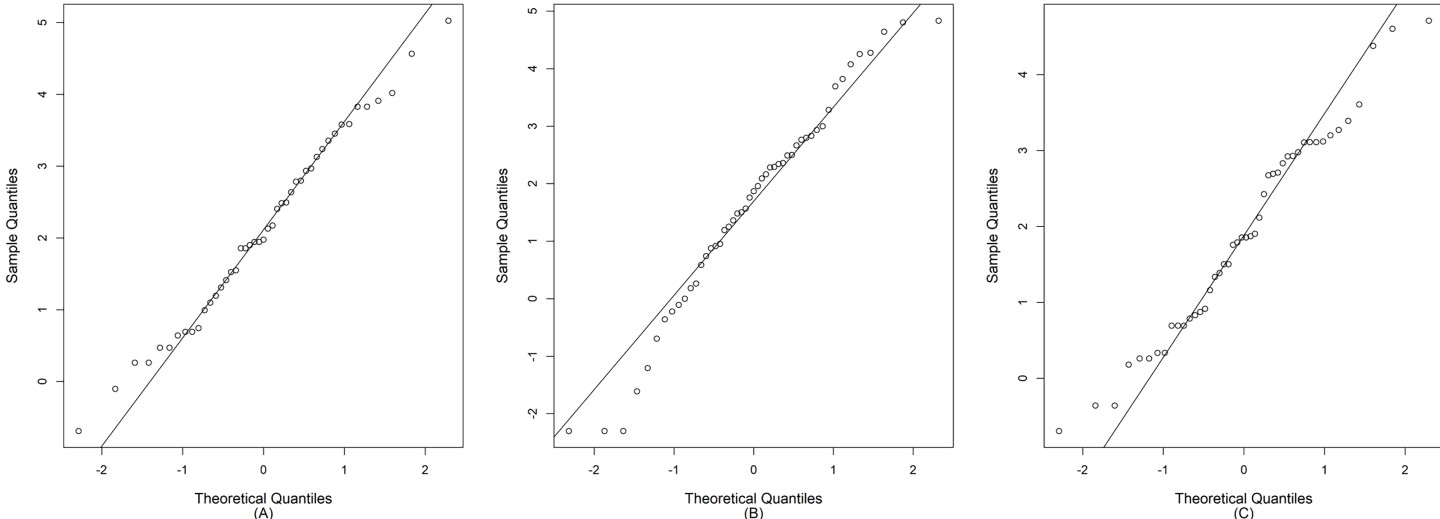

**Figure 2** The normal Q–Q plots of the log-transformation of the positive daily rainfall data from (A) Chiang Klang, (B) Tha Wang Pha, and (C) Pua in Nan, Thailand.

Since the CV is the ratio of the mean and the standard deviation of a population, it is free from units of measurement and is often used to measure the dispersion of data and compare it between populations. For statistical inference, several methods for constructing confidence intervals for the CV and functions of the CV have been suggested (*e.g. Pang et al., 2005*; *Hayter, 2015*; *Nam & Kwon, 2017*; *Yosboonruang, Niwitpong & Niwitpong, 2018*, *2019a*, *2019b*, *2020*; *Yosboonruang & Niwitpong, 2020*). However, since using the common CV of delta-lognormal distributions for statistical inference has not previously been reported, this has become our research interest as it is useful for measuring the dispersion in several independent data series, especially rainfall data.

Several statisticians have suggested confidence intervals for the common CV of normal and non-normal distributions. Once *Gupta, Ramakrishnan & Zhou (1999)* obtained the asymptotic variance of the common CV for normal distributions, they constructed confidence intervals and compared their coverage probabilities and expected lengths. *Tian (2005)* developed a method by using the concept of the generalized confidence interval (GCI) for the common CV. Subsequently, *Behboodian & Jafari (2008)* used the concept of generalized *p*-values and GCI to construct a new method and compared it with Tian's method (*Tian, 2005*); the former outperformed the others by attaining a suitable coverage probability and the shortest expected length. *Ng (2014)* constructed confidence intervals for the common CV of lognormal distributions using the generalized variables approach; the performance of the proposed method was similar to Tian's method (*Tian, 2005*). *Liu & Xu (2015)* constructed a confidence interval for the common CV of several normal populations based on the confidence distribution interval. *Thangjai & Niwitpong (2017)* proposed the adjusted method of variance estimates recovery (MOVER) to construct the confidence interval for the weighted CV of two-parameter exponential distributions; a performance comparison between it, GCI, and a large sample method revealed that GCI performed the best and adjusted MOVER was only suitable for data with

positive values. Recently, *Thangjai, Niwitpong & Niwitpong (2020a)* applied adjusted GCI and a computational method to construct confidence intervals for the common CV of normal distributions; they compared them with GCI and the adjusted MOVER, the results of which show that the adjusted GCI is appropriate for small samples and the computational method was suitable for large ones. In addition, *Thangjai, Niwitpong & Niwitpong (2020b)* extended the computational approach and MOVER to produce confidence intervals for the common CV of lognormal distributions and compared their performances with those employing fiducial GCI (FGCI) and Bayesian approaches, of which FGCI was the best. Unfortunately, the work of *Thangjai, Niwitpong & Niwitpong (2020b)* considered only the positively skewed distribution: lognormal distribution. In this work, we regarded the lognormal distribution that contained true zero values, the delta-lognormal distribution, for the confidence interval construction of the common CV. Therefore, the research by *Thangjai, Niwitpong & Niwitpong (2020b)* also needs to continue as rainfall data must be the delta-lognormal distribution.

The fact that daily rainfall data can usually be fitted to a delta-lognormal distribution after collecting data over a sufficiently long period has drawn interest from several researchers to present statistical inference for its parameters. Several researchers have suggested confidence intervals for the mean and functions of the mean of delta-lognormal distributions, such as the traditional method, the normal algorithm, the exponential algorithm (*Kvanli, Shen & Deng, 1998*), bootstrapping, the likelihood ratio, the signed log-likelihood ratio (*Zhou & Tu, 2000*; *Tian, 2005*; *Tian & Wu, 2006*), GCI (*Tian, 2005*; *Chen & Zhou, 2006*; *Li, Zhou & Tian, 2013*; *Wu & Hsieh, 2014*; *Hasan & Krishnamoorthy, 2018*; *Maneerat, Niwitpong & Niwitpong, 2018*, *2019b*), MOVER (*Maneerat, Niwitpong & Niwitpong, 2018*, *2019a*, *2019b*), Aitchison's estimator, a modified Cox's method, a modified Land's method, the profile likelihood interval (*Fletcher, 2008*; *Wu & Hsieh, 2014*), FGCI (*Li, Zhou & Tian, 2013*; *Hasan & Krishnamoorthy, 2018*; *Maneerat, Niwitpong & Niwitpong, 2019a*), as well as Bayesian approaches (*Maneerat, Niwitpong & Niwitpong, 2019a*). Moreover, confidence interval estimations for the variance (*Maneerat, Niwitpong & Niwitpong, 2020a*, *2020b*), CV (*Yosboonruang, Niwitpong & Niwitpong, 2018*, *2019a*, *2019b*), and functions of the CV (*Yosboonruang & Niwitpong, 2020*; *Yosboonruang & Niwitpong, 2020*) of delta-lognormal distributions have been suggested, including GCI, the modified Fletcher's method, FGCI, MOVER, the Bayesian approach, and bootstrapping.

Herein, we construct confidence intervals for the common CV of delta-lognormal distributions based on FGCI, the Bayesian approach, and MOVER. Their performances were compared in terms of their coverage probabilities and expected lengths. The methods for the confidence intervals estimation are presented in the next section. Subsequently, the results and discussion of a simulation study are analyzed, followed by the use of daily rainfall data to determine their applicability in real situations. Finally, conclusions on the study are offered.

## METHODS

Let $X_{ij}$, $i = 1, 2, \ldots, k$, $j = 1, 2, \ldots, n_i$ be a random variable of size $n_i$ from $k$ delta-lognormal distributions with density function

$$f\left(x_{ij}; \mu_i, \sigma_i^2, \delta_i\right) = (1 - \delta_i)I_0\left[x_{ij}\right] + \delta_i \frac{1}{x_{ij}\sqrt{2\pi}\sigma_i} \exp\left\{-\frac{1}{2}\left[\frac{\ln\left(x_{ij}\right) - \mu_i}{\sigma_i}\right]^2\right\} I_{(0,\infty)}\left[x_{ij}\right], \quad (1)$$

where $I_0\left[x_{ij}\right]$ is an indicator function for which the values are 1 when $x_{ij} = 0$, and 0 otherwise; $I_{(0,\infty)}\left[x_{ij}\right]$ are equal to 0 and 1 when $x_{ij} = 0$ and $x_{ij} > 0$, respectively; and $\delta_i = P\left(X_{ij} > 0\right)$. This distribution is a combination of lognormal and binomial distributions. The numbers of positive and zero observations are defined as $n_{i1}$ and $n_{i0}$, respectively, where $n_i = n_{i1} + n_{i0}$. According to *Aitchison (1955)*, the mean and variance of a delta-lognormal distribution are defined as

$$E\left(X_{ij}\right) = \delta_i \exp\left(\mu_i + \frac{\sigma_i^2}{2}\right) \tag{2}$$

and

$$\sigma_i^2 = \delta_i \exp\left(2\mu_i + \sigma_i^2\right)\left[\exp\left(\sigma_i^2\right) - \delta_i\right], \tag{3}$$

respectively. Since the CV computed from $\sigma_i/\mu_i$, then

$$CV(X_{ij}) = \eta_i = \left[\frac{\exp\left(\sigma_i^2\right) - \delta_i}{\delta_i}\right]^{\frac{1}{2}}. \tag{4}$$

By using the log-transformation (*Yosboonruang, Niwitpong & Niwitpong, 2018*), let

$$\varphi_i = \frac{1}{2}\left\{\ln\left[\exp\left(\sigma_i^2\right) - \delta_i\right] - \ln(\delta_i)\right\}. \tag{5}$$

The unbiased estimators for $\sigma_i^2$ and $\delta_i$ are $\hat{\sigma}_i^2 = \sum_{j=1}^{n_{i1}}\left[\ln\left(x_{ij}\right) - \hat{\mu}_i\right]^2/(n_{i1} - 1)$ and $\hat{\delta}_i = n_{i1}/n_i$, for $i = 1, 2, \ldots, k$, where $\hat{\mu}_i = \sum_{j=1}^{n_{i1}} \ln\left(x_{ij}\right)/n_{i1}$, respectively, then

$$\hat{\varphi}_i = \frac{1}{2}\left\{\ln\left[\exp\left(\hat{\sigma}_i^2\right) - \hat{\delta}_i\right] - \ln\left(\hat{\delta}_i\right)\right\}. \tag{6}$$

The approximately unbiased estimate variance of $\hat{\varphi}_i$ is

$$\hat{V}(\hat{\varphi}_i) \approx \frac{\left(\hat{b}_i - \hat{a}_i\right)\left(1 - \hat{a}_i\hat{b}_i\right) - n_{i1}(1 - \hat{a}_i)^2}{4n_{i1}(1 - \hat{a}_i)^2} + \frac{\hat{\sigma}_i^4}{2(n_{i1} - 1)}, \tag{7}$$

where $\hat{a}_i = \left(1 - \hat{\delta}_i\right)^{n_i - 1}$ and $\hat{b}_i = 1 + (n_i - 1)\hat{\delta}_i$. The ordinary form of the common log-transformed CV is given by

$$\tilde{\varphi} = \frac{\sum_{i=1}^{k} w_i\hat{\varphi}_i}{\sum_{i=1}^{k} w_i}, \tag{8}$$

where $w_i = 1/\hat{V}(\hat{\varphi}_i)$. Accordingly, the common CV is defined as

$$\tilde{\eta} = \exp(\tilde{\varphi}) = \exp\left(\frac{\sum_{i=1}^{k} w_i\hat{\varphi}_i}{\sum_{i=1}^{k} w_i}\right). \tag{9}$$

Here, the methods to establish the confidence intervals for the common CV for delta-lognormal distributions are provided in detail.

## FGCI

Let $X_{ij}$, $i = 1, 2, \ldots, k$, $j = 1, 2, \ldots, n_i$ be a random sample with density function $f(x_{ij}; \theta_i, \mu_i)$, where $\theta_i = (\delta_i, \sigma_i^2)$ are the parameters of interest and $\mu_i$ is a nuisance parameter. Let $x_{ij}$ be the observed values of $X_{ij}$. To construct the FGCI (*Weerahandi, 1993*; *Hannig, Iyer & Patterson, 2006*), the fiducial generalized pivotal quantity (FGPQ) $R(X_{ij}; x_{ij}, \theta_i, \mu_i)$ is needed to satisfy the following two properties:

1. For each $x_{ij}$, the conditional distribution of $R(X_{ij}; x_{ij}, \theta_i, \mu_i)$ is unaffected by the nuisance parameter.

2. The observed value of $R(X_{ij}; x_{ij}, \theta_i, \mu_i)$, $r(x_{ij}; x_{ij}, \theta_i, \mu_i)$, is the parameter of interest.

Given that $R_\alpha$ is the $100\alpha$−th percentile of $R(X_{ij}; x_{ij}, \theta_i, \mu_i)$, then $(R_{\alpha/2}, R_{1-\alpha/2})$ becomes the $100(1 - \alpha)\%$ two-sided FGCI for $\theta_i$. Hence, it is essential to use the FGPQs for $\delta_i$ and $\sigma_i^2$ to construct the confidence interval for the common CV $(\tilde{\eta})$ for delta-lognormal distributions.

Consider $k$ individual random samples $X_{i1}, X_{i2}, \ldots, X_{in_i}$. Following *Hannig (2009)* and *Li, Zhou & Tian (2013)*, the FGPQ for $\delta_i$ is as follows

$$R_{\delta_i} \sim \frac{1}{2} Beta(n_{i1}, n_{i0} + 1) + \frac{1}{2} Beta(n_{i1} + 1, n_{i0}). \tag{10}$$

Similarly, *Wu & Hsieh (2014)* followed the concept of *Krishnamoorthy & Mathew (2003)* to find the FGPQ for $\sigma_i^2$ defined as

$$R_{\sigma_i^2} = \frac{(n_{i1} - 1)\hat{\sigma}_i^2}{U_i}, \tag{11}$$

where $U_i \sim \chi_{n_{i1}-1}^2$. To find the FGPQ for $\hat{\varphi}$, we then substitute $R_{\delta_i}$ and $R_{\sigma_i^2}$ into Eq. (6) as follows:

$$R_{\hat{\varphi}_i} = \frac{1}{2} \left\{ \ln \left[ \exp \left( R_{\sigma_i^2} \right) - \ln(R_{\delta_i}) \right] - \ln(R_{\delta_i}) \right\}. \tag{12}$$

Consequently, the FGPQ for common CV $(\tilde{\eta})$ is

$$R_{\tilde{\eta}} = \exp \left( \frac{\sum_{i=1}^{k} R_{w_i} R_{\hat{\varphi}_i}}{\sum_{i=1}^{k} R_{w_i}} \right), \tag{13}$$

where the FGPQ for an estimated variance of $\hat{\varphi}_i$, for which $R_{w_i}$ is the inverse, is given by

$$R_{\hat{V}(\hat{\varphi}_i)} = \frac{(R_{b_i} - R_{a_i})(1 - R_{a_i} R_{b_i}) - n_{i1}(1 - R_{a_i})^2}{4n_{i1}(1 - R_{a_i})^2} + \frac{R_{\sigma_i^2}^2}{2(n_{i1} - 1)}, \tag{14}$$

where $R_{a_i} = (1 - R_{\delta_i})^{n_i - 1}$ and $R_{b_i} = 1 + (n_i - 1)R_{\delta_i}$.

Thus, we employ $R_{\tilde{\eta}}$ to produce the confidence interval for $\tilde{\eta}$. Consequently, the $100(1 - \alpha)\%$ two-sided confidence interval for $\tilde{\eta}$ based on FGCI becomes

$\left(R_{\bar{\eta}}(\alpha/2), R_{\bar{\eta}}(1 - \alpha/2)\right)$, which denote the $\alpha/2$th and $(1 - \alpha/2)$th percentiles of $R_{\bar{\eta}}$, respectively.

## Bayesian methods

Since random samples $X_{ij}$, $i = 1, 2, \ldots, k$, $j = 1, 2, \ldots, n_i$ have a delta-lognormal distribution with unknown parameters $\phi = \left(\delta_i^*, \mu_i, \sigma_i^2\right)$, where $\delta_i^* = 1 - \delta_i$, the likelihood function of $k$-individual random samples can be expressed as

$$L\left(\phi \mid x_{ij}\right) \propto \prod_{i=1}^{k} \left(\delta_i^*\right)^{n_{i0}} \delta_i^{n_{i1}} \left(\sigma_i^2\right)^{-\frac{n_{i1}}{2}} \exp\left\{-\frac{1}{2\sigma_i^2} \sum_{j=1}^{n_{i1}} \left[\ln\left(x_{ij}\right) - \mu_i\right]^2\right\}. \tag{15}$$

Subsequently, the Fisher information matrix of $\phi$ based on the partial derivatives of the log-likelihood functions for $\delta_i$, $\mu_i$, and $\sigma_i^2$ becomes

$$I(\phi) = \text{diag}\left[\frac{n_1}{\delta_1^* \delta_1} \quad \frac{n_1 \delta_1}{\sigma_1^2} \quad \frac{n_1 \delta_1}{2\left(\sigma_1^2\right)^2} \quad \cdots \quad \cdots \quad \cdots \quad \frac{n_k}{\delta_k^* \delta_k} \quad \frac{n_k \delta_k}{\sigma_k^2} \quad \frac{n_k \delta_k}{2\left(\sigma_k^2\right)^2}\right]. \tag{16}$$

In the present study, the Bayesian method is used to construct the equal-tailed confidence interval and the credible interval for the common CV. In the following section, we propose the independent Jeffreys and uniform priors.

### The Bayesian method using the independent Jeffreys prior

Since Jeffreys' prior for unknown parameter $\phi$ is derived from the square root of the determinant of Fisher information matrix $I(\phi)$, then $p(\phi) = \sqrt{|I(\phi)|}$. Since the parameters of interest $\vartheta = \left(\delta_i^*, \sigma_i^2\right)$, the independent Jeffreys prior for $\delta_i$ and $\sigma_i^2$ are $p(\delta_i) \propto \left(\delta_i^*\right)^{-\frac{1}{2}} \delta_i^{-\frac{1}{2}}$ and $p\left(\sigma_i^2\right) \propto 1/\sigma_i^2$ (*Harvey & van der Merwe, 2012*), respectively. Since $\delta_i^*$ and $\sigma_i^2$ are independent, then the independent Jeffreys prior for a delta-lognormal distribution is $p(\vartheta) \propto \prod_{i=1}^{k} \sigma_i^{-2} \left(\delta_i^*\right)^{-\frac{1}{2}} \delta_i^{-\frac{1}{2}}$. Thus, the joint posterior density of $\phi$ becomes

$$p\left(\phi \mid x_{ij}\right) = \prod_{i=1}^{k} \frac{1}{Beta\left(n_{i0} + \frac{1}{2}, n_{i1} + \frac{1}{2}\right)} \left(\delta_i^*\right)^{\left(n_{i0} + \frac{1}{2}\right) - 1} \delta_i^{\left(n_{i1} + \frac{1}{2}\right) - 1} \frac{1}{\sqrt{2\pi} \frac{\sigma_i}{\sqrt{n_{i1}}}} \exp\left[-\frac{1}{2\frac{\sigma_i^2}{n_{i1}}} (\mu_i - \hat{\mu}_i)^2\right]$$

$$\times \frac{\left[\frac{(n_{i1} - 1)\hat{\sigma}_i^2}{2}\right]^{\frac{n_{i1} - 1}{2}}}{\Gamma\left(\frac{n_{i1} - 1}{2}\right)} \left(\sigma_i^2\right)^{-\frac{n_{i1} - 1}{2} - 1} \exp\left[-\frac{\frac{(n_{i1} - 1)\hat{\sigma}_i^2}{2}}{\sigma_i^2}\right], \tag{17}$$

where $\hat{\mu}_i = \sum_{j=1}^{n_{i1}} \ln\left(x_{ij}\right)/n_{i1}$, and $\hat{\sigma}_i^2 = \sum_{j=1}^{n_{i1}} \left[\ln\left(x_{ij}\right) - \hat{\mu}_i\right]^2/(n_{i1} - 1)$. This leads to the posterior density of $\delta_i^*$ given by

$$p\left(\delta_i^* \mid x_{ij}\right) \propto \prod_{i=1}^{k} \frac{1}{Beta\left(n_{i0} + \frac{1}{2}, n_{i1} + \frac{1}{2}\right)} \left(\delta_i^*\right)^{\left(n_{i0} + \frac{1}{2}\right) - 1} \delta_i^{\left(n_{i1} + \frac{1}{2}\right) - 1}, \tag{18}$$

which is a beta distribution with parameters $n_{i0} + 1/2$ and $n_{i1} + 1/2$, denoted by $\delta_i^* \mid x_{ij} \sim Beta(n_{i0} + 1/2, n_{i1} + 1/2)$. Similarly, the posterior density of $\sigma_i^2$ can be derived as

$$p\left(\sigma_i^2 \mid x_{ij}\right) \propto \frac{\left[\frac{(n_{i1}-1)\hat{\sigma}_i^2}{2}\right]^{\frac{n_{i1}-1}{2}}}{\Gamma\left(\frac{n_{i1}-1}{2}\right)} \left(\sigma_i^2\right)^{-\frac{n_{i1}-1}{2}-1} \exp\left[-\frac{\frac{(n_{i1}-1)\hat{\sigma}_i^2}{2}}{\sigma_i^2}\right], \tag{19}$$

which is in the general form of an inverse gamma distribution denoted by $\sigma_i^2 \mid x_{ij} \sim Inv - Gamma\left[(n_{i1} - 1)/2, (n_{i1} - 1)\hat{\sigma}_i^2/2\right]$.

### The Bayesian method using the uniform prior

Because all possible values are equally likely *a priori* for the uniform prior, then it is a constant function of *a priori* probability (*Stone, 2013*; *O'Reilly & Mars, 2015*). According to *Bolstad & Curran (2016)*, the uniform prior for $\delta_i^*$ and $\sigma_i^2$ are proportional to 1, which can be defined as $p\left(\delta_i^*\right) \propto 1$ and $p\left(\sigma_i^2\right) \propto 1$, respectively. It is well-known that $\delta_i^*$ is independent of $\sigma_i^2$, thereby the uniform prior for the parameters of interest for a delta-lognormal distribution is $p\left(\delta_i^*, \sigma_i^2\right) \propto 1$. Accordingly, the joint posterior density function is defined as

$$p\left(\phi \mid x_{ij}\right) = \prod_{i=1}^{k} \frac{1}{Beta(n_{i0} + 1, n_{i1} + 1)} \left(\delta_i^*\right)^{n_{i0}} \delta_i^{n_{i1}} \frac{1}{\sqrt{2\pi} \frac{\sigma_i}{\sqrt{n_{i1}}}} \exp\left[-\frac{1}{2\frac{\sigma_i^2}{n_{i1}}} (\mu_i - \hat{\mu}_i)^2\right]$$

$$\times \frac{\left[\frac{(n_{i1}-2)\hat{\sigma}_i^2}{2}\right]^{\frac{n_{i1}-2}{2}}}{\Gamma\left(\frac{n_{i1}-2}{2}\right)} \left(\sigma_i^2\right)^{-\frac{n_{i1}-2}{2}-1} \exp\left[-\frac{\frac{(n_{i1}-2)\hat{\sigma}_i^2}{2}}{\sigma_i^2}\right], \tag{20}$$

where $\hat{\mu}_i = \sum_{j=1}^{n_{i1}} \ln(x_{ij})/n_{i1}$, and $\hat{\sigma}_i^2 = \sum_{j=1}^{n_{i1}} \left[\ln(x_{ij}) - \hat{\mu}_i\right]^2/(n_{i1} - 1)$. We can derived the posterior density of $\delta_i^*$ as

$$p\left(\delta_i^* \mid x_{ij}\right) \propto \prod_{i=1}^{k} \frac{1}{Beta(n_{i0} + 1, n_{i1} + 1)} \left(\delta_i^*\right)^{n_{i0}} \delta_i^{n_{i1}}, \tag{21}$$

which is consequently a density function of a beta distribution, *i.e.* $\delta_i^* \mid x_{ij} \sim Beta(n_{i0} + 1, n_{i1} + 1)$. For $\sigma_i^2$, the posterior density has an inverse gamma distribution with respective shape and scale parameters $(n_{i1} - 2)/2$ and $(n_{i1} - 2)\hat{\sigma}_i^2/2$ which expressed as

$$p\left(\sigma_i^2 \mid x_{ij}\right) \propto \prod_{i=1}^{k} \frac{\left(\frac{(n_{i1}-2)\hat{\sigma}_i^2}{2}\right)^{\frac{n_{i1}-2}{2}}}{\Gamma\left(\frac{n_{i1}-2}{2}\right)} \left(\sigma_i^2\right)^{-\frac{n_{i1}-2}{2}-1} \exp\left(-\frac{\frac{(n_{i1}-2)\hat{\sigma}_i^2}{2}}{\sigma_i^2}\right). \tag{22}$$

Subsequently, we construct the confidence intervals and the credible intervals for the common CV by substituting the posterior densities of $\delta_i^*$ and $\sigma_i^2$ from the independent Jeffreys and uniform priors into Eqs. ( 6), (7), and (9).

## MOVER

Following the method of *Zou & Donner (2008)*, let $\varphi_1$ and $\varphi_2$ be the parameter of interest and then let $\hat{\varphi}_1$ and $\hat{\varphi}_2$ be the independent estimators of $\varphi_1$ and $\varphi_2$, respectively. Furthermore, the lower and upper confidence limits for $\varphi_1 + \varphi_2$ are

$$CI_{\varphi_1+\varphi_2} = \left[L_{\varphi_1+\varphi_2}, U_{\varphi_1+\varphi_2}\right] = \hat{\varphi}_1 + \hat{\varphi}_2 \pm z_{\alpha/2}\sqrt{\widehat{Var}(\hat{\varphi}_1) + \widehat{Var}(\hat{\varphi}_2)}, \tag{23}$$

Subsequently, let $l_i$ and $u_i$, for $i = 1, 2$, be the lower and upper bounds of the confidence interval for $\varphi_i$, respectively. Since $l_i$ and $u_i$ provide the possible parameter values, then $l_1 + l_2$ is close to $L_{\varphi_1+\varphi_2}$ and $u_1 + u_2$ is close to $U_{\varphi_1+\varphi_2}$. To obtain the lower limit $L_{\varphi_1+\varphi_2}$, the estimated variance of $\hat{\varphi}_i$ at $\varphi_i = l_i$ is given by

$$\widehat{Var}(\hat{\varphi}_{l_i}) = \frac{(\hat{\varphi}_i - l_i)^2}{z_{\alpha/2}^2}. \tag{24}$$

Similarly, to obtain the upper limit $U_{\varphi_1+\varphi_2}$, the estimated variance of $\hat{\varphi}_i$ at $\varphi_i = u_i$ is given by

$$\widehat{Var}(\hat{\varphi}_{u_i}) = \frac{(u_i - \hat{\varphi}_i)^2}{z_{\alpha/2}^2}. \tag{25}$$

Next, by substituting $\widehat{Var}(\hat{\varphi}_{l_i})$ and $\widehat{Var}(\hat{\varphi}_{u_i})$ into Eq. ( 23), we obtain

$$L_{\varphi_1+\varphi_2} = \hat{\varphi}_1 + \hat{\varphi}_2 - \sqrt{(\hat{\varphi}_1 - l_1)^2 + (\hat{\varphi}_2 - l_2)^2} \tag{26}$$

and

$$U_{\varphi_1+\varphi_2} = \hat{\varphi}_1 + \hat{\varphi}_2 - \sqrt{(u_1 - \hat{\varphi}_1)^2 + (u_2 - \hat{\varphi}_2)^2}. \tag{27}$$

Thereby, the unbiased estimate variance of $\hat{\varphi}_i$ at $\varphi_i = l_i$ and $\varphi_i = u_i$ can be expressed as

$$\widehat{Var}(\hat{\varphi}_i) = \frac{1}{2}\left[\frac{(\hat{\varphi}_i - l_i)^2}{z_{\alpha/2}^2} + \frac{(u_i - \hat{\varphi}_i)^2}{z_{\alpha/2}^2}\right], i = 1, 2. \tag{28}$$

When this concept is extended to $k$ parameters, the lower and upper confidence limits for $\upsilon = \sum_{i=1}^{k} \varphi_i$ are given by

$$L_\upsilon = \upsilon - \sqrt{(\hat{\varphi}_1 - l_1)^2 + (\hat{\varphi}_2 - l_2)^2 + \ldots + (\hat{\varphi}_k - l_k)^2} \tag{29}$$

and

$$U_\upsilon = \upsilon + \sqrt{(u_1 - \hat{\varphi}_1)^2 + (u_2 - \hat{\varphi}_2)^2 + \ldots + (u_k - \hat{\varphi}_k)^2}. \tag{30}$$

According to *Krishnamoorthy & Oral (2017)* and recalling the common log-transformed CV from Eq. ( 8), the upper and lower confidence limits for $\sigma_i^2$ and $\delta_i$ are required to construct the confidence interval for the common CV of delta-lognormal distributions. Since the estimate of $\sigma_i^2$ is

$$\hat{\sigma}_i^2 = \frac{1}{n_{i1} - 1}\sum_{j=1}^{n_{i1}}\left[\ln(x_{ij}) - \hat{\mu}_i\right]^2, \tag{31}$$

where $(n_{i1} - 1)\hat{\sigma}_i^2/\sigma_i^2 \sim \chi_{n_{i1}-1}^2$, the $100(1 - \alpha)\%$ confidence interval for $\sigma_i^2$ is derived as

$$CI_{\sigma_i^2} = \left(l_{\sigma_i^2}, u_{\sigma_i^2}\right) = \left[\frac{(n_{i1}-1)\hat{\sigma}_i^2}{\chi_{1-\alpha/2, n_{i1}-1}^2}, \frac{(n_{i1}-1)\hat{\sigma}_i^2}{\chi_{\alpha/2, n_{i1}-1}^2}\right]. \tag{32}$$

To construct the confidence interval for $\delta_i$, the concept of the variance stabilizing transformation proposed by *DasGupta (2008)* and *Wu & Hsieh (2014)* was used. Therefore, the confidence interval for $\delta_i$ is given by

$$CI_{\delta_i} = (l_{\delta_i}, u_{\delta_i}) = \sin^2\left(\arcsin\sqrt{\hat{\delta}_i} \pm \frac{1}{2\sqrt{n_i}} Z_{i(1-\alpha/2)}\right). \tag{33}$$

Since $\varphi_i = \left\{\ln\left[\exp(\sigma_i^2) - \delta_i\right] - \ln(\delta_i)\right\}/2$, then let

$$l_i = \frac{1}{2}\left\{\ln\left[\exp\left(l_{\sigma_i^2}\right) - l_{\delta_i}\right] - \ln(l_{\delta_i})\right\} \tag{34}$$

and

$$u_i = \frac{1}{2}\left\{\ln\left[\exp\left(u_{\sigma_i^2}\right) - u_{\delta_i}\right] - \ln(u_{\delta_i})\right\}, \tag{35}$$

for $i = 1, 2, \ldots, k$. Therefore, the $100(1-\alpha)\%$ confidence interval for $\tilde{\eta}$ based on MOVER can be written as

$$CI_{\tilde{\eta}}^m = \left[L_{\tilde{\eta}}^m, U_{\tilde{\eta}}^m\right], \tag{36}$$

where

$$L_{\tilde{\eta}}^m = \exp\left(\tilde{\varphi} - \sqrt{\sum_{i=1}^k w_i^2(\hat{\varphi}_i - l_i)^2 \Big/ \sum_{i=1}^k w_i^2}\right) \tag{37}$$

and

$$U_{\tilde{\eta}}^m = \exp\left(\tilde{\varphi} + \sqrt{\sum_{i=1}^k w_i^2(u_i - \hat{\varphi}_i)^2 \Big/ \sum_{i=1}^k w_i^2}\right). \tag{38}$$

## RESULTS

### Monte Carlo simulation studies

The R statistical program was used to run the Monte Carlo simulation study and calculate the results for evaluating the performances of FGCI, MOVER, and the Bayesian intervals with the independent Jeffreys or uniform priors. The criteria for choosing the best performing confidence interval were coverage probability $\geq 0.95$ and the shortest expected length for each scenario tested. To generate the data, we set the number of populations as $k = 3, 5, 10$; sample sizes as $n_1 = n_2 = \ldots = n_k = n = 25, 50, 100$; probabilities of non-zero values as $\delta_1 = \delta_2 = \ldots = \delta_k = \delta = 0.2, 0.5, 0.8$; and variances as $\sigma_1^2 = \sigma_2^2 = \ldots = \sigma_k^2 = \sigma^2 = 0.1, 0.5, 1.0, 2.0$. For each combination of parameters, 10,000 simulation runs were generated together with 2,000 replications for FGCI and the Bayesian approaches by applying Algorithms 1 and 2, respectively.

**Algorithm 1**

(For $i$ = 1 to M)

    Generate $x_{ij}, i = 1, 2, \ldots, k, j = 1, 2, \ldots, n_i$ from a delta-lognormal distribution.

    Compute $\hat{\delta}_i$ and $\hat{\sigma}_i^2$.

    (For $j$ = 1 to K)

        Generate $U_i \sim \chi_{n_{i1}-1}^2$, $Beta(n_{i1}, n_{i0} + 1)$, and $Beta(n_{i1} + 1, n_{i0})$.

        Compute $R_{\sigma_i^2}$, $R_{\delta_i}$, $R_{\hat{\varphi}_i}$, and $R_{\tilde{\eta}}$.

    (End $j$ loop)

    Compute the $100(1 - \alpha/2)$ % confidence interval for $\tilde{\eta}$.

(End $i$ loop)

**Algorithm 2**

(For $i$ = 1 to M)

    Generate $x_{ij}, i = 1, 2, \ldots, k, \ j = 1, 2, \ldots, n_i$ from a delta-lognormal distribution.

    Compute $\hat{\delta}_i$ and $\hat{\sigma}_i^2$.

    (For $j$ = 1 to K)

        Generate the posterior densities of $\delta_i^* \mid x_{ij}$.

            1. Independent Jeffreys prior: $\delta_i^* \mid x_{ij} \sim Beta\left(n_{i0} + \frac{1}{2}, n_{i1} + \frac{1}{2}\right)$.

            2. Uniform prior: $\delta_i^* \mid x_{ij} \sim Beta(n_{i0} + 1, n_{i1} + 1)$.

        Generate the posterior densities of $\sigma_i^2 \mid x_{ij}$.

            1. Independent Jeffreys prior: $\sigma_i^2 \mid x_{ij} \sim Inv - Gamma\left[\frac{n_{i1}-1}{2}, \frac{(n_{i1}-1)\hat{\sigma}_i^2}{2}\right]$.

            2. Uniform prior: $\sigma_i^2 \mid x_{ij} \sim Inv - Gamma\left[\frac{n_{i1}-2}{2}, \frac{(n_{i1}-2)\hat{\sigma}_i^2}{2}\right]$.

        Compute $\hat{\varphi}_i$, $\hat{V}(\hat{\varphi}_i)$, and $\tilde{\eta}$.

    (End $j$ loop)

    Compute the $100(1 - \alpha/2)$ % confidence intervals and credible intervals for $\tilde{\eta}$.

(End $i$ loop)

**Algorithm 3**

(For $i$ = 1 to M)

    Generate $x_{ij}, i = 1, 2, \ldots, k, j = 1, 2, \ldots, n_i$ from a delta-lognormal distribution.

    Compute $\hat{\delta}_i$ and $\hat{\sigma}_i^2$.

    Compute $\hat{\varphi}_i$ and $\hat{V}(\hat{\varphi}_i)$.

    Compute $l_{\sigma_i^2}, u_{\sigma_i^2}, l_{\delta_i}, u_{\delta_i}, l_i, u_i$.

    Compute the $100(1 - \alpha/2)$ % confidence interval for $\tilde{\eta}$.

(End $i$ loop)

**Table 1 The results for the 95% two-sided confidence intervals for the common CV of delta-lognormal distributions for $k = 3$.**

| $n_i$ | $\delta_i$ | $\sigma_i^2$ | Coverage probabilities (Expected lengths) | | | | | |
|---|---|---|---|---|---|---|---|---|
| | | | **FGCI** | **E-B.Indj** | **E-B.Uni** | **C-B.Indj** | **C-B.Uni** | **MOVER** |
| 25 | 0.5 | 0.1 | **0.9740** | **0.9537** | **0.9601** | 0.9357 | 0.9411 | 0.2536 |
| | | | (0.7272) | (0.7291) | **(0.7249)** | (0.7079) | (0.7037) | (0.3519) |
| | | 0.5 | **0.9983** | **0.9689** | **0.9734** | **0.9518** | **0.9564** | 0.5541 |
| | | | (4.2363) | (1.5221) | (1.6132) | **(1.3548)** | (1.4144) | (0.4478) |
| | | 1.0 | **0.9775** | **0.9613** | **0.9672** | 0.9429 | **0.9518** | 0.7872 |
| | | | (28.1984) | **(4.2206)** | (4.7049) | (3.2428) | (3.5018) | (1.4839) |
| | | 2.0 | 0.9027 | **0.9528** | **0.9600** | 0.9408 | 0.9485 | 0.9133 |
| | | | (7.4009) | **(34.3846)** | (42.9115) | (18.1504) | (20.8831) | (8.0816) |
| | 0.8 | 0.1 | **0.9959** | **0.9676** | **0.9823** | 0.9479 | **0.9699** | 0.1611 |
| | | | (0.5921) | (0.4309) | (0.4381) | (0.4188) | **(0.4263)** | (0.1540) |
| | | 0.5 | **0.9994** | **0.9658** | **0.9700** | **0.9575** | **0.9687** | 0.7894 |
| | | | (9.1319) | (0.8848) | (0.9205) | **(0.8193)** | (0.8486) | (0.3732) |
| | | 1.0 | **0.9890** | **0.9563** | **0.9623** | **0.9526** | **0.9589** | 0.9445 |
| | | | (274.3619) | (2.0822) | (2.1950) | **(1.7915)** | (1.8716) | (1.1456) |
| | | 2.0 | 0.9204 | **0.9518** | **0.9567** | 0.9464 | **0.9548** | 0.9705 |
| | | | (13.9706) | (9.4530) | (10.2449) | (6.8908) | **(7.3329)** | (4.7949) |
| 50 | 0.2 | 0.1 | **0.9510** | **0.9611** | **0.9554** | 0.9424 | 0.9346 | 0.5551 |
| | | | **(1.2188)** | (1.3347) | (1.3067) | (1.2849) | (1.2573) | (0.7675) |
| | | 0.5 | **0.9940** | **0.9734** | **0.9721** | **0.9589** | **0.9580** | 0.4001 |
| | | | (5.3666) | (3.0387) | (3.3163) | **(2.5773)** | (2.7137) | (0.7416) |
| | | 1.0 | **0.9668** | **0.9663** | **0.9700** | 0.9436 | 0.9473 | 0.6903 |
| | | | (61.6887) | **(10.2196)** | (12.7965) | (6.8382) | (7.7720) | (2.3985) |
| | | 2.0 | 0.8941 | **0.9573** | **0.9674** | 0.9403 | 0.9473 | 0.8591 |
| | | | (10.2166) | **(205.9432)** | (470.3159) | (61.6888) | (89.7961) | (16.4537) |
| | 0.5 | 0.1 | **0.9746** | **0.9573** | **0.9592** | 0.9454 | 0.9463 | 0.7691 |
| | | | **(0.4985)** | (0.5278) | (0.5246) | (0.5178) | (0.5146) | (0.3383) |
| | | 0.5 | **0.9991** | **0.9665** | **0.9689** | **0.9517** | **0.9533** | 0.4400 |
| | | | (2.3261) | (0.9184) | (0.9322) | **(0.8743)** | (0.8851) | (0.2156) |
| | | 1.0 | **0.9977** | **0.9554** | **0.9608** | 0.9435 | 0.9462 | 0.7862 |
| | | | (27.7099) | **(2.0097)** | (2.0670) | (1.8046) | (1.8466) | (0.7810) |
| | | 2.0 | 0.9383 | 0.9498 | **0.9545** | 0.9385 | 0.9434 | 0.9348 |
| | | | (24.2678) | (8.0467) | **(8.3999)** | (6.4058) | (6.6170) | (3.8741) |
| | 0.8 | 0.1 | **0.9960** | **0.9675** | **0.9766** | **0.9514** | **0.9658** | 0.3313 |
| | | | (0.3738) | (0.3069) | (0.3088) | **(0.3013)** | (0.3033) | (0.1317) |
| | | 0.5 | **0.9997** | **0.9607** | **0.9665** | **0.9507** | **0.9593** | 0.7357 |
| | | | (1.6860) | (0.5521) | (0.5606) | **(0.5316)** | (0.5397) | (0.2003) |
| | | 1.0 | **0.9980** | **0.9563** | **0.9584** | 0.9458 | **0.9513** | **0.9516** |
| | | | (5.1781) | (1.1695) | (1.1911) | (1.0905) | (1.1088) | **(0.7058)** |
| | | 2.0 | 0.9464 | **0.9521** | **0.9554** | 0.9483 | **0.9515** | 0.9824 |

| Table 1 (continued) | | | | | | | | |
|---|---|---|---|---|---|---|---|---|
| $n_i$ | $\delta_i$ | $\sigma_i^2$ | Coverage probabilities (Expected lengths) | | | | | |
| | | | FGCI | E-B.Indj | E-B.Uni | C-B.Indj | C-B.Uni | MOVER |
| | | | (293.3686) | (3.9878) | (4.0900) | (3.4771) | (3.5501) | **(2.7229)** |
| 100 | 0.2 | 0.1 | 0.9461 | **0.9591** | **0.9537** | 0.9434 | 0.9352 | 0.8726 |
| | | | (0.8134) | (0.9378) | **(0.9241)** | (0.9166) | (0.9035) | (0.6757) |
| | | 0.5 | **0.9965** | 0.9691 | 0.9679 | **0.9508** | 0.9470 | 0.2665 |
| | | | (3.4026) | **(1.6493)** | (1.6684) | (1.5542) | (1.5659) | (0.3647) |
| | | 1.0 | **0.9947** | 0.9613 | 0.9631 | 0.9433 | 0.9420 | 0.6284 |
| | | | (13.0351) | **(3.7143)** | (3.8315) | (3.2441) | (3.3177) | (1.0735) |
| | | 2.0 | 0.9326 | **0.9547** | 0.9601 | 0.9384 | 0.9410 | 0.8846 |
| | | | (181.1185) | **(16.6849)** | (17.6932) | (12.4128) | (12.9191) | (6.1118) |
| 100 | 0.5 | 0.1 | 0.9685 | 0.9578 | 0.9585 | 0.9445 | 0.9462 | 0.8992 |
| | | | **(0.3491)** | (0.3777) | (0.3764) | (0.3724) | (0.3710) | (0.2704) |
| | | 0.5 | **0.9996** | 0.9604 | 0.9632 | 0.9495 | **0.9501** | 0.3157 |
| | | | (1.3697) | (0.6091) | (0.6121) | (0.5933) | **(0.5961)** | (0.1058) |
| | | 1.0 | **0.9990** | 0.9553 | 0.9574 | 0.9379 | 0.9415 | 0.7982 |
| | | | (3.7435) | **(1.2257)** | (1.2378) | (1.1620) | (1.1725) | (0.4857) |
| | | 2.0 | **0.9668** | 0.9545 | 0.9548 | 0.9442 | 0.9475 | **0.9549** |
| | | | (32.9162) | (4.0100) | (4.0691) | (3.6039) | (3.6477) | **(2.4343)** |
| | 0.8 | 0.1 | **0.9950** | 0.9581 | 0.9676 | 0.9450 | **0.9559** | 0.5995 |
| | | | (0.2532) | (0.2194) | (0.2200) | (0.2164) | **(0.2170)** | (0.1167) |
| | | 0.5 | 1.0000 | **0.9562** | 0.9595 | 0.9465 | **0.9503** | 0.7093 |
| | | | (1.0189) | (0.3729) | (0.3754) | (0.3648) | **(0.3672)** | (0.1204) |
| | | 1.0 | **0.9992** | 0.9537 | 0.9562 | 0.9443 | 0.9465 | **0.9560** |
| | | | (2.3875) | (0.7571) | (0.7631) | (0.7289) | (0.7344) | **(0.4796)** |
| | | 2.0 | **0.9662** | 0.9496 | **0.9522** | 0.9445 | 0.9479 | **0.9872** |
| | | | (16.9898) | (2.3221) | (2.3448) | (2.1655) | (2.1851) | **(1.7740)** |

Note:
E-B.Indj and E-B.Uni represented the respective equal-tailed Bayesian intervals based on independent Jeffreys and uniform priors, and C-B.Indj and C-B.Uni represented the respective Bayesian credible intervals based on independent Jeffrey's and uniform priors. Bold indicates the coverage probability ≥ 0.95 and the shortest expected length.

The result for the 95% confidence and credible intervals for the common CV of delta-lognormal distributions for various sample sizes, probabilities of non-zero values, and variances are reported in Tables 1–3 and displayed in Figs. 3–8. The equal-tailed Bayesian credible intervals based on the independent Jeffreys or uniform priors produced coverage probabilities close to or greater than the nominal confidence level for almost all of the scenarios whereas the others could achieve this in only some of them. Furthermore, in terms of the expected lengths, the equal-tailed based on independent Jeffreys prior were shorter than the uniform prior for all cases. In addition, the expected lengths of the Bayesian credible interval based on the independent Jeffreys prior were shorter than the others in almost every case when $\sigma_i^2 = 0.5$. For all $k$ and sample sizes together with $\sigma_i^2 = 1, 2$, the expected lengths of the equal-tailed Bayesian based on the independent Jeffreys prior were the shortest when $\delta_i = 0.2, 0.5$, while MOVER had the shortest

**Table 2 The results for the 95% two-sided confidence intervals for the common CV of delta-lognormal distributions for $k = 5$.**

| $n_i$ | $\delta_i$ | $\sigma_i^2$ | Coverage probabilities (Expected lengths) | | | | | |
|---|---|---|---|---|---|---|---|---|
| | | | **FGCI** | **E-B.Indj** | **E-B.Uni** | **C-B.Indj** | **C-B.Uni** | **MOVER** |
| 25 | 0.5 | 0.1 | **0.9834** | **0.9596** | **0.9646** | 0.9374 | 0.9438 | 0.1197 |
| | | | (0.7505) | (0.7303) | **(0.7259)** | (0.7088) | (0.7045) | (0.3301) |
| | | 0.5 | **0.9997** | **0.9710** | **0.9746** | **0.9531** | 0.9595 | 0.4830 |
| | | | (3.7038) | (1.5339) | (1.6273) | **(1.3637)** | (1.4246) | (0.3499) |
| | | 1.0 | **0.9737** | **0.9600** | **0.9647** | 0.9423 | **0.9506** | 0.7917 |
| | | | (8.9088) | (4.2573) | (4.7602) | (3.2717) | **(3.5364)** | (1.2034) |
| | | 2.0 | 0.8603 | **0.9541** | **0.9613** | 0.9410 | 0.9483 | 0.9096 |
| | | | (1,933.3646) | **(32.2422)** | (40.4745) | (17.2785) | (19.8882) | (5.9389) |
| | 0.8 | 0.1 | **0.9997** | **0.9697** | **0.9840** | 0.9485 | **0.9730** | 0.0701 |
| | | | (0.6148) | (0.4309) | (0.4382) | (0.4187) | **(0.4265)** | (0.1438) |
| | | 0.5 | 1.0000 | **0.9639** | **0.9713** | **0.9574** | **0.9671** | 0.8460 |
| | | | (3.4724) | (0.8808) | (0.9159) | **(0.8160)** | (0.8447) | (0.3396) |
| | | 1.0 | **0.9836** | **0.9583** | **0.9637** | **0.9510** | **0.9614** | **0.9642** |
| | | | (10.7410) | (2.0830) | (2.1955) | (1.7916) | (1.8719) | **(1.0345)** |
| | | 2.0 | 0.8901 | **0.9517** | **0.9576** | 0.9493 | **0.9560** | **0.9807** |
| | | | (3.67E+04) | (9.7354) | (10.5885) | (7.0602) | (7.5199) | **(4.1883)** |
| 50 | 0.2 | 0.1 | **0.9532** | **0.9604** | **0.9526** | 0.9401 | 0.9310 | 0.4326 |
| | | | **(1.2400)** | (1.3288) | (1.3007) | (1.2798) | (1.2516) | (0.7301) |
| | | 0.5 | **0.9987** | **0.9743** | **0.9730** | **0.9556** | **0.9546** | 0.2693 |
| | | | (5.1024) | (3.0276) | (3.2924) | **(2.5680)** | (2.7012) | (0.5583) |
| | | 1.0 | **0.9647** | **0.9655** | **0.9690** | 0.9441 | 0.9467 | 0.6413 |
| | | | (16.6022) | **(10.0192)** | (12.3389) | (6.7304) | (7.6160) | (1.6963) |
| | | 2.0 | 0.8562 | **0.9539** | **0.9636** | 0.9407 | 0.9498 | 0.8377 |
| | | | (3.08E+05) | **(187.9496)** | (362.9737) | (60.4077) | (85.9882) | (10.2873) |
| | 0.5 | 0.1 | **0.9803** | **0.9609** | **0.9630** | 0.9457 | **0.9501** | 0.7854 |
| | | | (0.5169) | (0.5283) | (0.5251) | (0.5183) | **(0.5151)** | (0.3322) |
| | | 0.5 | 1.0000 | **0.9661** | **0.9688** | **0.9519** | **0.9550** | 0.3610 |
| | | | (2.4601) | (0.9163) | (0.9303) | **(0.8721)** | (0.8834) | (0.1694) |
| | | 1.0 | **0.9990** | **0.9582** | **0.9620** | 0.9446 | 0.9489 | 0.8114 |
| | | | (6.0419) | **(2.0201)** | (2.0770) | (1.8123) | (1.8540) | (0.6894) |
| | | 2.0 | 0.9276 | **0.9529** | **0.9570** | 0.9440 | 0.9463 | 0.9459 |
| | | | (1,419.0445) | **(7.9109)** | (8.2496) | (6.2988) | (6.5140) | (3.4257) |
| | 0.8 | 0.1 | **0.9996** | **0.9640** | **0.9743** | 0.9492 | **0.9641** | 0.2056 |
| | | | (0.3905) | (0.3067) | (0.3087) | (0.3012) | **(0.3032)** | (0.1238) |
| | | 0.5 | 1.0000 | **0.9650** | **0.9703** | **0.9546** | **0.9618** | 0.8014 |
| | | | (1.8753) | (0.5542) | (0.5624) | (0.5337) | **(0.5412)** | (0.1868) |
| | | 1.0 | **0.9993** | **0.9541** | **0.9547** | 0.9456 | **0.9501** | **0.9760** |
| | | | (4.2360) | (1.1716) | (1.1949) | (1.0922) | (1.1119) | **(0.6724)** |
| | | 2.0 | 0.9321 | **0.9552** | **0.9576** | **0.9518** | **0.9555** | **0.9932** |

| Table 2 (continued) | | | | | | | | |
|---|---|---|---|---|---|---|---|---|
| $n_i$ | $\delta_i$ | $\sigma_i^2$ | Coverage probabilities (Expected lengths) | | | | | |
| | | | FGCI | E-B.Indj | E-B.Uni | C-B.Indj | C-B.Uni | MOVER |
| | | | (45.8789) | (4.0252) | (4.1253) | (3.5054) | (3.5809) | **(2.5923)** |
| 100 | 0.2 | 0.1 | **0.9507** | **0.9586** | **0.9514** | 0.9442 | 0.9368 | 0.9115 |
| | | | **(0.8347)** | (0.9364) | (0.9220) | (0.9150) | (0.9014) | (0.6683) |
| | | 0.5 | 0.9995 | 0.9667 | 0.9640 | 0.9484 | 0.9459 | 0.1510 |
| | | | (3.6099) | (1.6424) | **(1.6607)** | (1.5490) | (1.5595) | (0.2957) |
| | | 1.0 | 0.9980 | 0.9626 | 0.9638 | 0.9438 | 0.9443 | 0.6174 |
| | | | (8.7078) | **(3.7265)** | (3.8501) | (3.2539) | (3.3293) | (0.8938) |
| | | 2.0 | 0.9178 | 0.9578 | 0.9612 | 0.9435 | 0.9464 | 0.8842 |
| | | | (1,302.6130) | **(16.4958)** | (17.4678) | (12.2904) | (12.7883) | (5.0720) |
| 100 | 0.5 | 0.1 | 0.9707 | 0.9559 | 0.9573 | 0.9452 | 0.9448 | 0.9455 |
| | | | **(0.3598)** | (0.3784) | (0.3768) | (0.3730) | (0.3715) | (0.2692) |
| | | 0.5 | 1.0000 | 0.9618 | 0.9621 | 0.9494 | 0.9494 | 0.2473 |
| | | | (1.5439) | **(0.6091)** | (0.6120) | (0.5933) | (0.5958) | (0.0838) |
| | | 1.0 | 1.0000 | 0.9541 | 0.9577 | 0.9425 | 0.9459 | 0.8244 |
| | | | (3.9566) | **(1.2291)** | (1.2421) | (1.1653) | (1.1763) | (0.4456) |
| | | 2.0 | 0.9672 | 0.9541 | 0.9558 | 0.9444 | 0.9464 | 0.9717 |
| | | | (15.8155) | (4.0068) | (4.0614) | (3.6001) | (3.6428) | (2.2992) |
| | 0.8 | 0.1 | 0.9985 | 0.9622 | 0.9688 | 0.9499 | 0.9598 | 0.5759 |
| | | | (0.2646) | (0.2194) | (0.2200) | (0.2164) | **(0.2171)** | (0.1142) |
| | | 0.5 | 1.0000 | 0.9607 | 0.9647 | 0.9483 | 0.9549 | 0.7699 |
| | | | (1.1485) | (0.3738) | (0.3762) | (0.3656) | **(0.3679)** | (0.1127) |
| | | 1.0 | 1.0000 | 0.9499 | 0.9530 | 0.9430 | 0.9461 | 0.9857 |
| | | | (2.6290) | (0.7549) | (0.7615) | (0.7270) | (0.7329) | **(0.4680)** |
| | | 2.0 | 0.9598 | 0.9516 | 0.9523 | 0.9466 | 0.9489 | 0.9969 |
| | | | (6.1687) | (2.3383) | (2.3604) | (2.1803) | (2.1988) | **(1.7304)** |

Note:
Bold indicates the coverage probability ≥ 0.95 and the shortest expected length.

expected lengths for $\delta_i = 0.8$. For FGCI, the coverage probabilities and their expected lengths were very wide for all cases in which it is not reasonable for the construction of confidence interval. However, the equal-tailed Bayesian interval based on the independent Jeffreys prior can be used to derive the confidence interval for the common CV of delta-lognormal distributions since it produced coverage probabilities ≥ 0.95 for almost all cases, although the expected lengths were not always shorter than the other methods in some cases.

## Application of the methods to real datasets

Datasets of daily rainfall from Chiang Klang, Tha Wang Pha, and Pua in Nan, Thailand, were obtained from the Upper Northern Region Irrigation Hydrology Center. The reason for using these datasets is discussed previously. AIC and BIC were used to test the possible distributions of these datasets in which the non-zero observations follow a

Table 3 The results for the 95% two-sided confidence intervals for the common CV of delta-lognormal distributions for $k = 10$.

| $n_i$ | $\delta_i$ | $\sigma_i^2$ | Coverage probabilities (Expected lengths) | | | | | |
|---|---|---|---|---|---|---|---|---|
| | | | FGCI | E-B.Indj | E-B.Uni | C-B.Indj | C-B.Uni | MOVER |
| 25 | 0.5 | 0.1 | **0.9750** | **0.9578** | 0.9648 | 0.9353 | 0.9420 | 0.0262 |
| | | | (0.7387) | (0.7284) | **(0.7247)** | (0.7070) | (0.7032) | (0.3056) |
| | | 0.5 | **0.9999** | **0.9682** | 0.9716 | **0.9500** | **0.9550** | 0.3282 |
| | | | (3.7907) | (1.5242) | (1.6165) | **(1.3565)** | (1.4164) | (0.2387) |
| | | 1.0 | **0.9721** | **0.9568** | 0.9646 | 0.9396 | 0.9486 | 0.7670 |
| | | | (7.9902) | **(4.2284)** | (4.7050) | (3.2447) | (3.5022) | (0.9681) |
| | | 2.0 | 0.8170 | **0.9544** | 0.9617 | 0.9399 | 0.9486 | 0.9124 |
| | | | (27.2845) | **(35.0401)** | (46.9413) | (18.2674) | (21.6147) | (4.5799) |
| | 0.8 | 0.1 | 1.0000 | **0.9695** | 0.9829 | **0.9511** | **0.9734** | 0.0118 |
| | | | (0.5753) | (0.4324) | (0.4395) | **(0.4201)** | (0.4275) | (0.1331) |
| | | 0.5 | 1.0000 | **0.9658** | 0.9731 | **0.9566** | **0.9677** | 0.9021 |
| | | | (3.4758) | (0.8807) | (0.9150) | **(0.8155)** | (0.8441) | (0.3032) |
| | | 1.0 | **0.9796** | **0.9538** | 0.9594 | 0.9494 | **0.9584** | **0.9804** |
| | | | (7.0148) | (2.0894) | (2.2045) | (1.7966) | (1.8782) | **(0.9335)** |
| | | 2.0 | 0.8306 | **0.9552** | 0.9601 | 0.9488 | **0.9557** | **0.9900** |
| | | | (37.3845) | (9.6166) | (10.4834) | (6.9970) | (7.4504) | **(3.6911)** |
| 50 | 0.2 | 0.1 | 0.9490 | **0.9603** | 0.9553 | 0.9430 | 0.9356 | 0.2560 |
| | | | (1.2475) | (1.3345) | **(1.3065)** | (1.2848) | (1.2569) | (0.6806) |
| | | 0.5 | **0.9979** | **0.9723** | 0.9724 | **0.9541** | **0.9518** | 0.1051 |
| | | | (5.3392) | (3.0554) | (3.3326) | **(2.5815)** | (2.7157) | (0.3897) |
| | | 1.0 | **0.9650** | **0.9636** | 0.9685 | 0.9442 | 0.9463 | 0.5255 |
| | | | (11.0842) | **(10.4725)** | (12.9382) | (6.9200) | (7.8607) | (1.1454) |
| | | 2.0 | 0.8124 | **0.9575** | 0.9658 | 0.9442 | **0.9522** | 0.8009 |
| | | | (53.4270) | (1,142.6720) | (3,369.2176) | (122.0329) | **(222.4721)** | (6.6526) |
| | 0.5 | 0.1 | **0.9700** | **0.9601** | 0.9632 | 0.9453 | 0.9495 | 0.7897 |
| | | | **(0.5115)** | (0.5278) | (0.5248) | (0.5177) | (0.5149) | (0.3240) |
| | | 0.5 | 1.0000 | **0.9672** | 0.9702 | **0.9516** | **0.9553** | 0.2112 |
| | | | (2.6159) | (0.9159) | (0.9302) | **(0.8720)** | (0.8828) | (0.1163) |
| | | 1.0 | **0.9970** | **0.9596** | 0.9621 | 0.9458 | **0.9507** | 0.8171 |
| | | | (6.1760) | (2.0213) | (2.0769) | (1.8134) | **(1.8535)** | (0.6006) |
| | | 2.0 | 0.8956 | **0.9540** | 0.9584 | 0.9415 | 0.9459 | **0.9652** |
| | | | (15.6125) | (8.0203) | (8.3825) | (6.3810) | (6.5924) | **(3.0276)** |
| | 0.8 | 0.1 | **0.9976** | **0.9652** | 0.9745 | **0.9507** | **0.9651** | 0.0741 |
| | | | (0.3688) | (0.3067) | (0.3088) | **(0.3012)** | (0.3033) | (0.1153) |
| | | 0.5 | **0.9596** | **0.9557** | 0.9487 | 0.9433 | 0.8688 | 0.9300 |
| | | | (2.0073) | **(0.5524)** | (0.5606) | (0.5322) | (0.5395) | (0.1726) |
| | | 1.0 | **0.9975** | **0.9554** | 0.9598 | 0.9451 | **0.9516** | **0.9910** |
| | | | (4.2894) | (1.1691) | (1.1913) | (1.0899) | (1.1089) | **(0.6371)** |
| | | 2.0 | 0.8936 | **0.9554** | 0.9579 | **0.9513** | **0.9566** | **0.9992** |

| $n_i$ | $\delta_i$ | $\sigma_i^2$ | Coverage probabilities (Expected lengths) | | | | | |
|---|---|---|---|---|---|---|---|---|
| | | | FGCI | E-B.Indj | E-B.Uni | C-B.Indj | C-B.Uni | MOVER |
| | | | (7.8557) | (4.0458) | (4.1465) | (3.5224) | (3.5975) | **(2.4638)** |
| 100 | 0.2 | 0.1 | 0.9445 | **0.9577** | **0.9534** | 0.9437 | 0.9349 | **0.9551** |
| | | | (0.8343) | (0.9358) | (0.9215) | (0.9147) | (0.9008) | **(0.6605)** |
| | | 0.5 | **0.9996** | **0.9680** | **0.9669** | **0.9550** | 0.9487 | 0.0384 |
| | | | (3.8651) | (1.6467) | (1.6645) | **(1.5521)** | (1.5622) | (0.2242) |
| | | 1.0 | **0.9942** | **0.9614** | **0.9623** | 0.9459 | 0.9445 | 0.5322 |
| | | | (8.9324) | **(3.7215)** | (3.8417) | (3.2516) | (3.3245) | (0.6967) |
| | | 2.0 | 0.8966 | **0.9539** | **0.9579** | 0.9389 | 0.9412 | 0.8751 |
| | | | (22.0831) | **(16.7659)** | (17.8464) | (12.4326) | (12.9750) | (4.1159) |
| 100 | 0.5 | 0.1 | **0.9612** | **0.9547** | **0.9562** | 0.9431 | 0.9439 | **0.9816** |
| | | | (0.3580) | (0.3777) | (0.3764) | (0.3724) | (0.3711) | **(0.2680)** |
| | | 0.5 | 1.0000 | **0.9623** | **0.9624** | **0.9512** | **0.9519** | 0.1266 |
| | | | (1.6621) | (0.6085) | (0.6115) | **(0.5927)** | (0.5954) | (0.0594) |
| | | 1.0 | 1.0000 | **0.9561** | **0.9582** | 0.9433 | 0.9462 | 0.8441 |
| | | | (4.1570) | **(1.2303)** | (1.2424) | (1.1663) | (1.1766) | (0.4028) |
| | | 2.0 | **0.9565** | 0.9497 | **0.9523** | 0.9404 | 0.9409 | **0.9899** |
| | | | (8.2323) | (4.0172) | (4.0756) | (3.6098) | (3.6542) | **(2.1795)** |
| | 0.8 | 0.1 | **0.9923** | **0.9598** | **0.9679** | 0.9494 | **0.9589** | 0.5202 |
| | | | (0.2599) | (0.2192) | (0.2198) | (0.2163) | **(0.2168)** | (0.1112) |
| | | 0.5 | 1.0000 | **0.9590** | **0.9631** | **0.9501** | **0.9547** | 0.8216 |
| | | | (1.2451) | (0.3745) | (0.3770) | **(0.3663)** | (0.3686) | (0.1055) |
| | | 1.0 | 1.0000 | **0.9551** | **0.9568** | 0.9476 | 0.9483 | **0.9975** |
| | | | (2.7509) | (0.7554) | (0.7609) | (0.7273) | (0.7324) | **(0.4582)** |
| | | 2.0 | **0.9518** | 0.9497 | **0.9526** | 0.9469 | 0.9480 | **0.9999** |
| | | | (4.0077) | (2.3383) | (2.3610) | (2.1800) | (2.1998) | **(1.6955)** |

**Note:**
Bold indicates the coverage probability ≥ 0.95 and the shortest expected length.

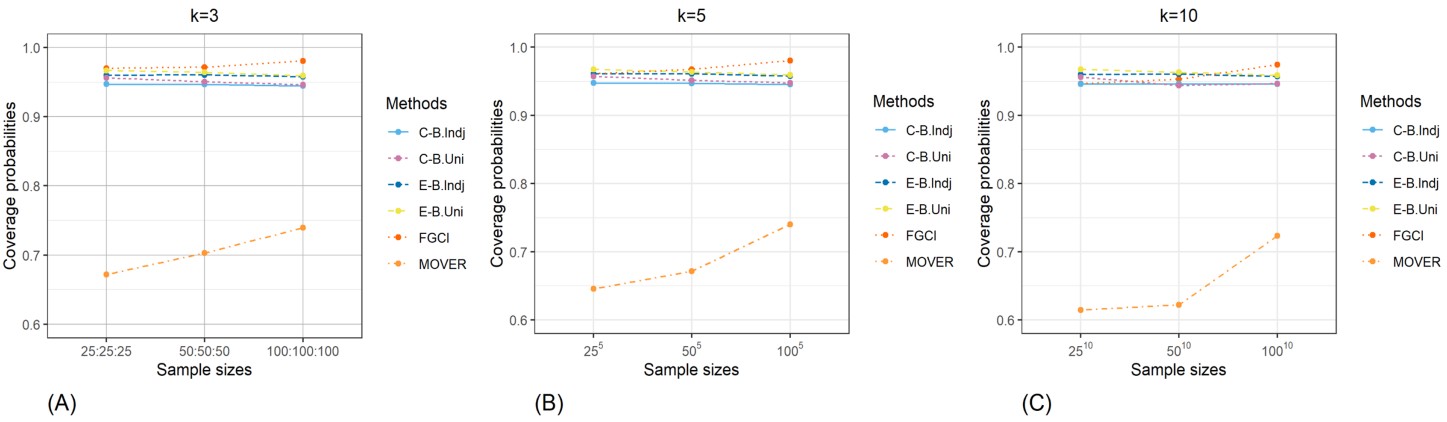

**Figure 3 Comparison of the coverage probabilities of the proposed methods according to sample sizes for (A) $k = 3$ (B) $k = 5$ (C) $k = 10$.**

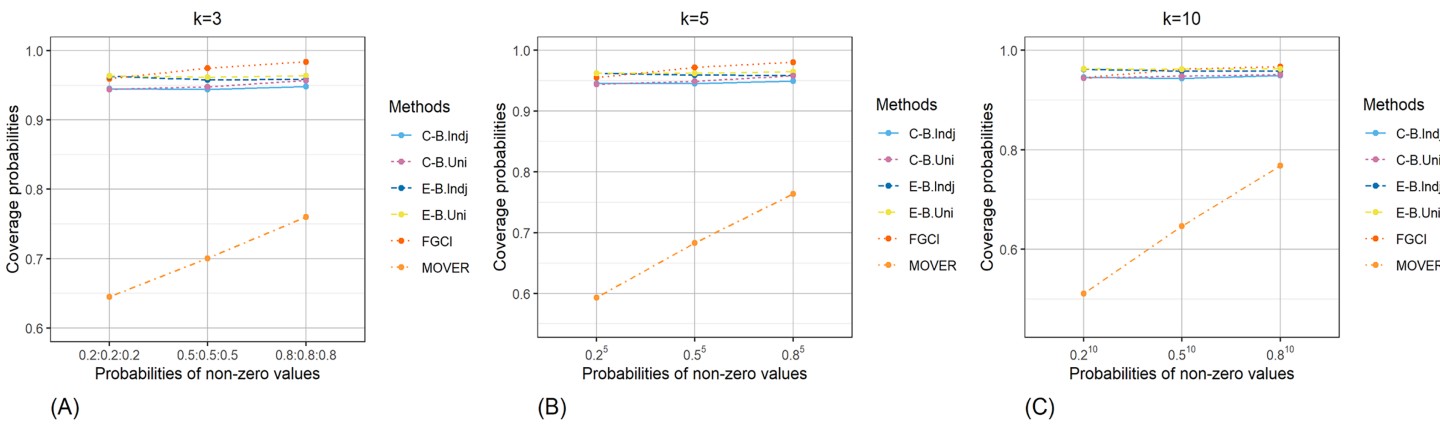

**Figure 4 Comparison of the coverage probabilities of the proposed methods according to probabilities of non-zero values for (A) $k = 3$ (B) $k = 5$ (C) $k = 10$.**

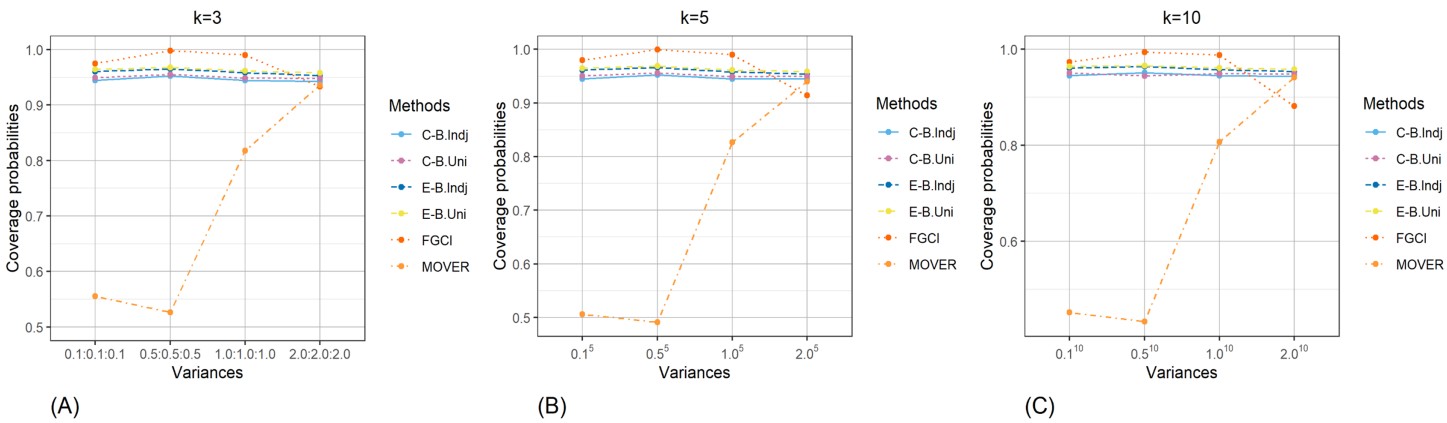

**Figure 5 Comparison of the coverage probabilities of the proposed methods according to variances for (A) $k = 3$ (B) $k = 5$ (C) $k = 10$.**

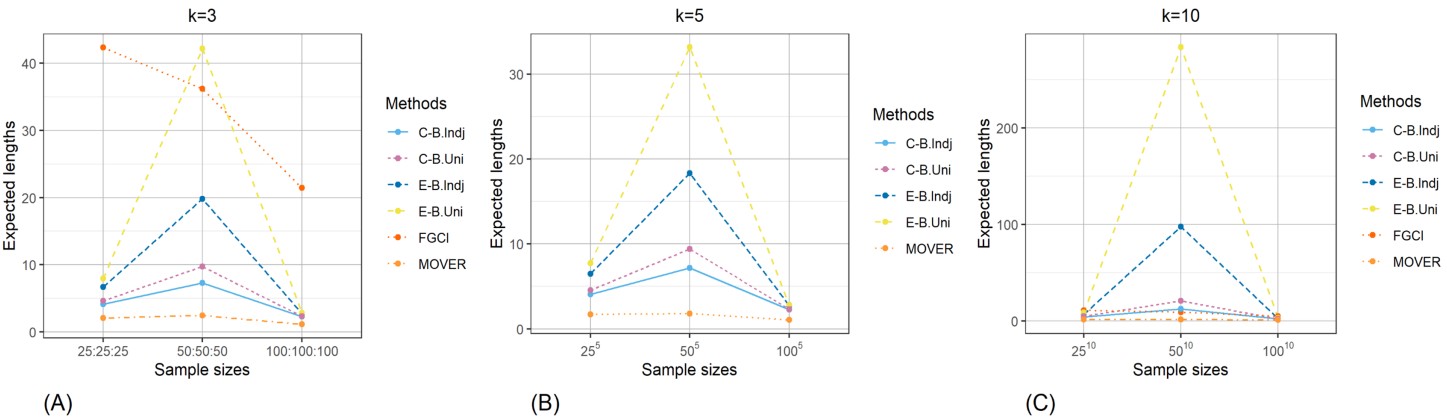

**Figure 6 Comparison of the expected lengths of the proposed methods according to sample sizes for (A) $k = 3$ (B) $k = 5$ (C) $k = 10$.**

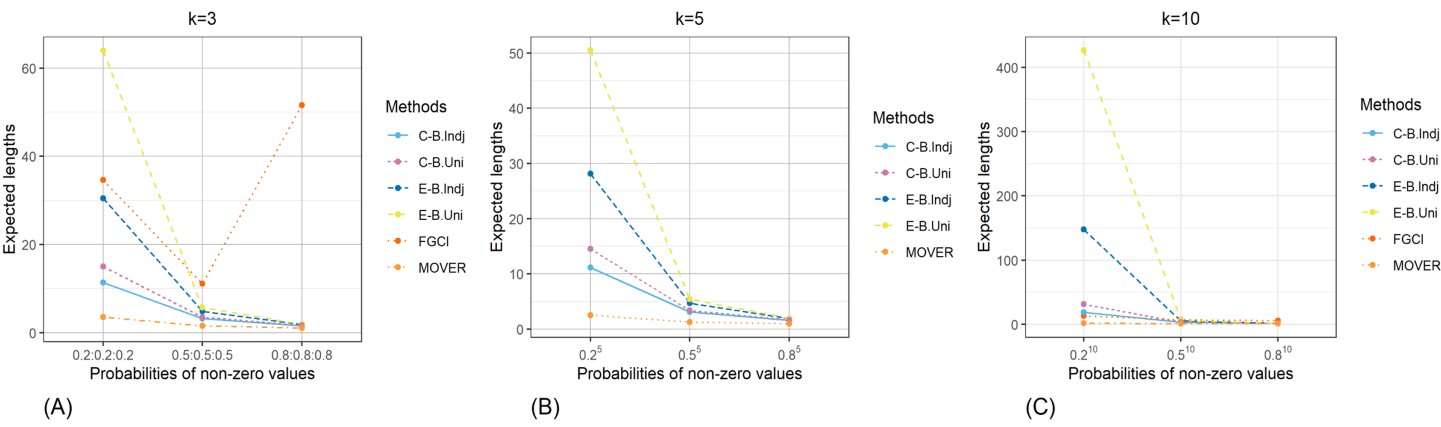

**Figure 7** Comparison of the expected lengths of the proposed methods according to probabilities of non-zero values for (A) $k = 3$ (B) $k = 5$ (C) $k = 10$.

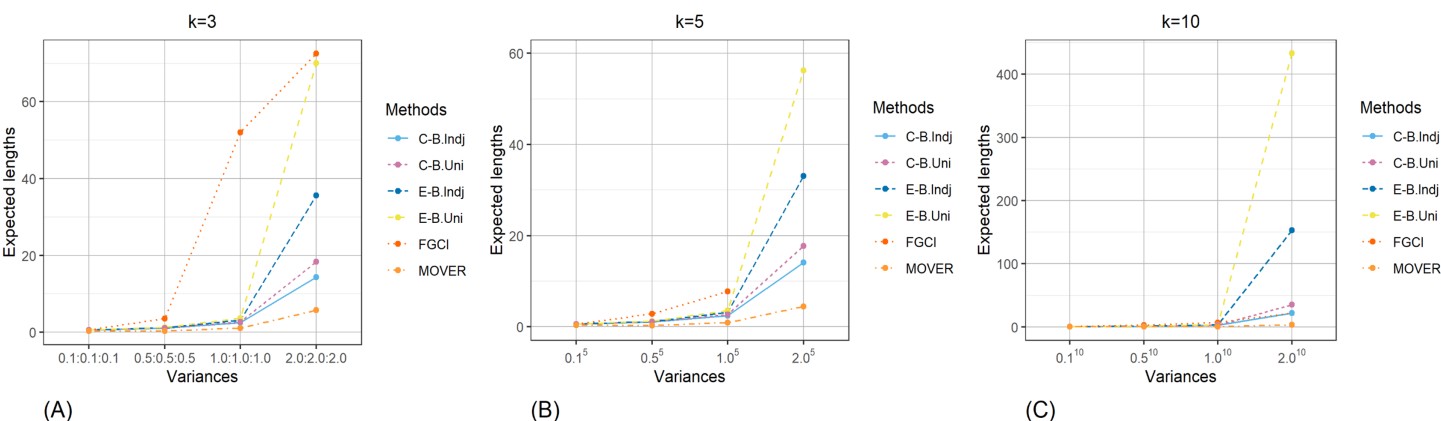

**Figure 8** Comparison of the expected lengths of the proposed methods according to variances for (A) $k = 3$ (B) $k = 5$ (C) $k = 10$.

right-skewed distribution (Tables 4 and 5, respectively). The results indicate that the non-zero observations in the three datasets most closely follow a lognormal distribution. Furthermore, the normal Q–Q plots *via* the log-transformation of non-zero observations shown in Fig. 2 reveal that they follow normal distributions. By testing the non-zero observations together with the binomial distributions of the true zero observations indicate that the daily rainfall data from the three areas follow delta-lognormal distributions.

The summary statistics for the three daily rainfall datasets were $n_1 : n_2 : n_3 = 62 : 62 : 62$; $\hat{\delta}_1 : \hat{\delta}_2 : \hat{\delta}_3 = 0.7258 : 0.7903 : 0.7419$; $\hat{\mu}_1 : \hat{\mu}_2 : \hat{\mu}_3 = 2.1189 : 1.6448 : 1.8971$; $\hat{\sigma}_1^2 : \hat{\sigma}_2^2 : \hat{\sigma}_3^2 = 1.7857 : 3.4406 : 1.8346$; and $\tilde{\eta} = 3.2011$. The 95% confidence and credible intervals for the common CV of the three daily rainfall datasets are summarized in Table 6. The results reveal that the three confidence intervals tested contained the real value of the parameter, thereby reinforcing the conclusions based on the simulation study results. However, the expected length of FGCI was the shortest, thereby making it a good choice for estimating the common CV in the dispersion of precipitation from the three areas in Nan province, Thailand.

**Table 4  The AIC values of the non-zero observations from Chiang Klang, Tha Wang Pha, and Pua in Nan, Thailand.**

| Areas | Distributions | | | | | |
|---|---|---|---|---|---|---|
| | Normal | Lognormal | Cauchy | Exponential | Gamma | Weibull |
| Chiang Klang | 430.7372 | **347.4835** | 387.5394 | 356.1853 | 355.0152 | 353.0018 |
| Tha Wang Pha | 477.9087 | **363.7785** | 415.4823 | 386.6203 | 366.9576 | 363.8529 |
| Pua | 425.2069 | **335.9760** | 379.7269 | 346.6206 | 344.5560 | 342.1551 |

Note:
Bold indicates the minimum AIC.

**Table 5  The BIC values of the non-zero observations from Chiang Klang, Tha Wang Pha, and Pua in Nan, Thailand.**

| Areas | Distributions | | | | | |
|---|---|---|---|---|---|---|
| | Normal | Lognormal | Cauchy | Exponential | Gamma | Weibull |
| Chiang Klang | 434.3506 | **351.0968** | 391.1527 | 357.9920 | 358.6286 | 356.6151 |
| Tha Wang Pha | 481.6923 | **367.5621** | 419.2659 | 388.5121 | 370.7412 | 367.6365 |
| Pua | 428.8642 | **339.6333** | 383.3842 | 348.4493 | 348.2133 | 345.8124 |

Note:
Bold indicates as the minimum BIC.

**Table 6  The 95% confidence intervals and credible intervals for the common CV of daily rainfall datasets from Chiang Klang, Tha Wang Pha, and Pua in Nan, Thailand.**

| Methods | Lower | Upper | Lengths |
|---|---|---|---|
| FGCI | 2.0363 | 4.4528 | 2.4165 |
| E-B.Indj | 1.8975 | 4.7920 | 2.8945 |
| E-B.U | 1.9039 | 5.0631 | 3.1592 |
| C-B.Indj | 1.7583 | 4.3644 | 2.6061 |
| C-B.U | 1.7540 | 4.4703 | 2.7163 |
| MOVER | 2.3642 | 5.1399 | 2.7757 |

# DISCUSSION

We extended the idea of *Thangjai, Niwitpong & Niwitpong (2020b)* who established confidence intervals using FGCI for the common CV of lognormal distributions to the context of the same distribution but with excess zeros. We then applied it to examine the dispersion in three daily rainfall datasets. In our case, the findings from the simulation study infer that the Bayesian methods were preferable to FGCI and MOVER for almost all cases. Although MOVER performed well for cases with a high proportion of non-zero values, it produced coverage probabilities that were lower than the nominal confidence level for cases with small variances. This is probably because the lower and upper bounds of the zero values are used in the confidence interval construction, and the combined effect with the other parameters caused the inadequate coverage probability results.

## CONCLUSION

Confidence intervals for the common CV of delta-lognormal distribution were constructed based on FGCI, two equal-tailed Bayesian credible intervals using the independent Jeffreys or uniform priors, and MOVER. Their coverage probabilities and expected lengths under various simulation scenarios were used to assess their efficacies. The equal-tailed Bayesian credible interval based on the independent Jeffreys prior provided superior coverage probabilities compared to other methods. Moreover, the equal-tailed Bayesian based on the uniform prior can be used as an alternative. This is due to the parameter to be estimated relying on the posterior densities of $\delta_i^*$ and $\sigma_i^2$. According to the results of the simulation study and the real data example, the Bayesian credible interval based on the independent Jeffreys prior is suitable for cases with small variances since it provided the narrowest length of the interval due to it falling in the domain of its posterior density. Furthermore, MOVER is the best choice when the proportion of non-zero values is high and the variance is large.

### Funding

This research was funded by Thailand Science Research and Innovation Fund, and King Mongkut's University of Technology North Bangkok (Contract no. KMUTNB-BasicR-64-26). The funders had no role in study design, data collection and analysis, decision to publish, or preparation of the manuscript.

### Grant Disclosures

The following grant information was disclosed by the authors:
Thailand Science Research and Innovation Fund.
King Mongkut's University of Technology North Bangkok: KMUTNB-BasicR-64-26.

### Competing Interests

The authors declare that they have no competing interests.

### Author Contributions

- Noppadon Yosboonruang conceived and designed the experiments, performed the experiments, analyzed the data, prepared figures and/or tables, authored or reviewed drafts of the paper, and approved the final draft.
- Sa-Aat Niwitpong conceived and designed the experiments, performed the experiments, authored or reviewed drafts of the paper, and approved the final draft.
- Suparat Niwitpong performed the experiments, analyzed the data, prepared figures and/or tables, authored or reviewed drafts of the paper, and approved the final draft.

### Data Availability

The raw data and R codes are available in the Supplemental Files.

## Supplemental Information

Supplemental information for this article can be found online at http://dx.doi.org/10.7717/peerj.12858#supplemental-information.

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
