# Peer review of "Bayesian computation for the common coefficient of variation of delta-lognormal distributions with application to common rainfall dispersion in Thailand"

_PeerJ, doi:10.7717/peerj.12858_

## Round 0.1 · original submission · Major Revisions

We now have received three review reports on your manuscript. I have considered them, and based on the advice from the reviewers I think that your manuscript needs major revision according to the review comments. In particular, you should pay more attention to the comments from reviewer 2 while revising your manuscript.

Reviewer 1 ·

Basic reporting

Minor. Lines 15, 43, etc., describe rainfall as following a delta lognormal distribution. The authors consider daily rainfall. Add “daily” throughout (e.g., replacement at line 81).

Experimental design

No comment

Validity of the findings

My sole major concern is that the authors’ Monte-Carlo simulations were made for the delta log-normal. The authors find good performance with the equal-tailed Bayesian based on the independent Jeffreys prior (Figures 1-4). For that analysis type, please add additional simulation with small discrepancy to the probability distribution and evaluate coverage under such conditions.

Additional comments

No comment

Reviewer 2 ·

Basic reporting

The paper investigates the estimations for the common coefficient of variation of delta-lognormal distributions. The Bayesian parametric estimators are derived. They compare the new estimator with three existing estimators by using simulations and real data(rainfall data in Thailand) analysis. The results show that the equal-tailed Bayesian based on the independent Jeffreys prior was suitable.
It is interesting to give the Bayesian confidence intervals for the common coefficient of variation of delta-lognormal distributions. The results are reasonable and correct.

Experimental design

no comment

Validity of the findings

The emphasis of this article is not confirmed,It should be an applied one, just from the title.In this case,It should start with the actual issue(rainfall data) to illustrate the importance of interval estimation.From the existing literature,the rainfall data follow the delta-lognormal distribution,while, the existing confirm interval estimates are not available based on this data,it necessary to study the estimation based on the Lognormal distribution.Then,the next issue is the difficulty of direct estimation?The reason to take Bayes? At last,conclusion with estimate data from simulation & data. The reasonable illustrate of the research methodologies is not available,just copy & paste.

Reviewer 3 ·

Basic reporting

The paper is interesting. However, the motivation should be given for more details and discussion section must be provided.

Experimental design

See the attached file

Validity of the findings

The methods used in this work is quite similar to the paper of the authors published in PeerJ in 2019. This work uses population group k > 2.

Additional comments

Please see the attached file

Annotated reviews are not available for download in order to protect the identity of reviewers who chose to remain anonymous.

---

## Round 0.2 · Minor Revisions

Your manuscript still needs minor changes before it can be officially accepted for publication.

Reviewer 1 ·

Basic reporting

No comment

Experimental design

No comment

Validity of the findings

In my previous review, I wrote: My sole major concern is that the authors’ Monte-Carlo simulations were made for the delta log-normal. The authors find good performance with the equal-tailed Bayesian based on the independent Jeffreys prior (Figures 1-4). For that analysis type, please add additional simulation with small discrepancy to the probability distribution and evaluate coverage under such conditions.

The authors added sigma^2 = 0.1 to Table 1 and Table 2. This was the addition of small variances. That is not what I requested. The problem is that the authors assumed delta log-normal and then did simulations with delta log-normal. Figure 2 are important but not persuasive. To simulate "discrepancy to the probability distribution" please generate delta log-normal as currently, and then add additional noise additive (mean 0) or multiplicative (mean 1), as the authors think best, and assure that the coverage performance shown in Tables 1 and 2 do not deteriorate substantially.

Additional comments

No comment

Reviewer 3 ·

Basic reporting

All my concerns are corrected.

Experimental design

-

Validity of the findings

-

Additional comments

-

---

## Author Rebuttal · Round 0.2

**Response to the Reviewers**

| | |
|---|---|
| **Journal:** | PeerJ |
| **Manuscript ID:** | 63996 |
| **Title name:** | Bayesian computation for the common coefficient of variation of delta-lognormal distributions with application to common rainfall dispersion in Thailand |
| **Authors:** | Noppadon Yosboonruang, Sa-Aat Niwitpong, and Suparat Niwitpong |

**Dear Reviewers**,

We are very grateful for your constructive comments to our manuscript. The manuscript was revised in accordance with your valuable suggestions. Below are our point-by-point replies to your comments.

**Lists of correction:**

**Reviewer 1**

1. Minor. Lines 15, 43, etc., describe rainfall as following a delta lognormal distribution. The authors consider daily rainfall. Add "daily" throughout (e.g., replacement at line 81).
     **Response:**
     Thank you for your suggestion. We added daily rainfall instead of rainfall according to the context of each sentence throughout the revised manuscript.

2. My sole major concern is that the authors' Monte-Carlo simulations were made for the delta log-normal. The authors find good performance with the equal-tailed Bayesian based on the independent Jeffreys prior (Figures 1-4). For that analysis type, please add additional simulation with small discrepancy to the probability distribution and evaluate coverage under such conditions.
     **Response:**
     We added additional simulation with small discrepancy ($\sigma^2 = 0.1$) to the probability distribution and evaluate coverage under such conditions which presented in Tables 1-3 and Figures 3-8 in a revised manuscript.

**Reviewer 2**

1. The emphasis of this article is not confirmed, it should be an applied one, just from the title. In this case, it should start with the actual issue (rainfall data) to illustrate the importance of interval estimation.

    **Response:**

    Thank you for your recommendation. We have made improvements starting with rainfall data to illustrate the importance of interval estimation in the first paragraph of introduction section on pages 1-2 in a revised manuscript.

2. From the existing literature, the rainfall data follow the delta-lognormal distribution, while the existing confirm interval estimates are not available based on this data, it necessary to study the estimation based on the Lognormal distribution. Then, the next issue is the difficulty of direct estimation?

    **Response:**

    Estimating the coefficient of variation of the delta-lognormal distribution, equation (4), is more complicated than the lognormal distribution ($\delta_i = 0$). Therefore, it is difficult to directly estimate the common coefficient of variation of the delta-lognormal distributions because zero values must be estimated from the binomial distributions.

3. The reason to take Bayes?

    **Response:**

    We proposed Bayesian method because the highest posterior density interval by this method is the shortest possible interval containing $100(1-\alpha)$ % of the posterior probability such that the density within the interval have a higher probability than that outside. Moreover, we can give the posterior density function as determined by likelihood function and the several priors' distributions.

4. Conclusion with estimate data from simulation & data. The reasonable illustrate of the research methodologies is not available, just copy & paste.

    **Response:**

    We have summarized the results of the actual data in lines 224-227 on page 15 in a revised manuscript as follows:

    "The results reveal that the three confidence intervals tested contained the real value of the parameter, thereby reinforcing the conclusions based on the simulation study results. However, the expected length of FGCI was the shortest, thereby making it a good choice for estimating the common CV in the dispersion of precipitation from the three areas in Nan province, Thailand."

    Moreover, we explained further in the conclusion section in lines 244-248 on page 16 in a revised manuscript as follows:

    "The results indicate that… This is due to the parameter to be estimated relying on the posterior densities of $\delta_i^*$ and $\sigma_i^2$. According to the results of the simulation study and

the real data example, the Bayesian credible interval based on the independent Jeffreys prior is suitable for cases with small variances since it provided the narrowest length of the interval due to it falling in the domain of its posterior density."
* * *
**Reviewer 3**

1. What does the word "common" mean for this work? Please describe.
   **Response:**
   The word "common" for this work means mutually or joint which is used to study more than one delta-lognormal coefficients of variation at one time.

2. What is the method you used to derive the estimated variance in (7)? How?
   **Response:**
   We used the delta method to derive the estimated variance in (7) which can be seen from the article by Yosboonruang et al., 2018.
   "Yosboonruang, N., Niwitpong, S.-A., and Niwitpong, S. (2018). Confidence intervals for the coefficient of variation of the delta-lognormal distribution. In Anh, L. H., Dong, L. S., Kreinovich, V., and Thach, N. N., editors, Econometrics for Financial Applications, volume 760 of Studies in Computational Intelligence, pages 327–337. Springer International Publishing, Cham."

3. Please give a discussion why the MOVER method provides low performance. Especially, for large n, as the method depends on CLT.
   **Response:**
   We discussed about the MOVER method provides low performance in the discussion section in lines 233-237 on page 16 in a revised manuscript as follows:
   "Although MOVER performed well for cases with a high proportion of non-zero values, it produced coverage probabilities that were lower than the nominal confidence level for cases with small variances. This is probably because the lower and upper bounds of the zero values are used in the confidence interval construction, and the combined effect with the other parameters caused the inadequate coverage probability results.".

4. Your results for the expected length are not consistent (Figure 4).
   **Response:**
   For Figure 4 (changed to Figure 6 in a revised manuscript), we ignored FGCI method since it is too wide of the expected length.

5. It will be useful if you could provide the R code and a set of real data for application, not only the code for simulation.
   **Response:**
   We provided the R code and a set of real data for application as supplemental file "R-code-raw data.R".

6. The data used here are interested, but what is the reason to use. I would expect to see the other type of data. So, it will be different from "Measuring the dispersion of rainfall using Bayesian confidence intervals.... (2019)".

   **Response:**

   The rainfall data was used because it consists of positive values and true zero values. The positive values have lognormal distribution and true zero values have binomial distribution which corresponds to the delta-lognormal distribution. Moreover, Thailand has flooding in monsoon season due to heavy rain in some areas causing damage. Therefore, we interested to study the dispersion of rainfall in Thailand to apply in planning for preventing with such incidents that may occur in the future. Since this manuscript focused on daily rainfall data. Therefore, the other type of data will be presented here. In addition, we added an example of medical charges data which taken from Zhou and Tu (1999).

   "Zhou, X. H. and Tu, W. (1999). Comparison of several independent population means when their samples contain log-normal and possibly zero observations. Biometrics, 55:645–651."

   The medical charges were divided into two inpatient groups including control group (non-treatment) and intervention-based treatment group. The summary statistics are $n_1 = 147$, $\hat{\delta}_1 = 0.19$, $\hat{\mu}_1 = 9.03$, $\hat{\sigma}_1^2 = 1.37$ and $n_2 = 119$, $\hat{\delta}_2 = 0.17$, $\hat{\mu}_2 = 9.33$, $\hat{\sigma}_2^2 = 0.41$. The 95% confidence intervals and credible intervals for the common coefficient of variation of medical care costs data as follows:

   | Methods | Lower | Upper | Lengths |
   |---------|-------|-------|---------|
   | FGCI | 2.7465 | 4.5308 | 1.7843 |
   | E-B.Indj | 3.1924 | 8.3646 | 5.1722 |
   | E-B.U | 3.1533 | 8.2896 | 5.1363 |
   | C-B.Indj | 2.9653 | 7.4937 | 4.5284 |
   | C-B.U | 2.8734 | 7.4031 | 4.5297 |
   | MOVER | 2.7328 | 3.1510 | 0.4182 |

   This result indicated that MOVER based method outperform the others. Moreover, it might be useful for determining inpatient care costs.

7. Please also fit your data to other distributions which are in the right skew. A measure, for example, AIC should be provided.

   **Response:**

   We fitted the daily rainfall data to other distributions which are in the right skew as follows.

[Figure]

Figure 1. The Cauchy Q-Q plots of the positive daily rainfall data from (A) Chiang Klang
(B) Tha Wang Pha, and (C) Pua in Nan, Thailand.

[Figure]

Figure 2. The exponential Q-Q plots of the positive daily rainfall data from (A) Chiang
Klang (B) Tha Wang Pha, and (C) Pua in Nan, Thailand.

[Figure]

Figure 3. The gamma Q-Q plots of the positive daily rainfall data from (A) Chiang Klang
(B) Tha Wang Pha, and (C) Pua in Nan, Thailand.

[Figure]

Figure 4. The Weibull Q-Q plots of the positive daily rainfall data from (A) Chiang Klang (B) Tha Wang Pha, and (C) Pua in Nan, Thailand.

The measure, for example, AIC and BIC are

$$AIC = 2k - 2\ln(L)$$

and

$$BIC = k\ln(n) - 2\ln(L),$$

where $k$ is the number of estimated parameters in the model, $n$ is the number of observations, and $L$ is the maximized value of the likelihood function for the model.

Best Regards,
Noppadon Yosboonruang, Sa-Aat Niwitpong, and Suparat Niwitpong
The authors

---

## Round 0.3 · accepted · Accept

I am pleased to inform you that the current version of your manuscript has been accepted to publish by PeerJ.